# Gut Microbiota and Complications of Type-2 Diabetes

**DOI:** 10.3390/nu14010166

**Published:** 2021-12-30

**Authors:** Camelia Oana Iatcu, Aimee Steen, Mihai Covasa

**Affiliations:** 1College of Medicine and Biological Sciences, Stefan cel Mare University of Suceava, 720229 Suceava, Romania; oana.iatcu@usm.ro; 2College of Medicine, “Grigore T. Popa” University of Medicine and Pharmacy, 700115 Iasi, Romania; 3Department of Basic Medical Sciences, College of Osteopathic Medicine, Western University of Health Sciences, Pomona, CA 91766, USA; aimee.steen@westernu.edu

**Keywords:** gut microbiota dysbiosis, diabetes complications, retinopathies, nephropathies, microvascular complications, macrovascular complications

## Abstract

The gut microbiota has been linked to the emergence of obesity, metabolic syndrome and the onset of type 2 diabetes through decreased glucose tolerance and insulin resistance. Uncontrolled diabetes can lead to serious health consequences such as impaired kidney function, blindness, stroke, myocardial infarction and lower limb amputation. Despite a variety of treatments currently available, cases of diabetes and resulting complications are on the rise. One promising new approach to diabetes focuses on modulating the gut microbiota with probiotics, prebiotics, synbiotics and fecal microbial transplantation. Differences in gut microbiota composition have been observed in preclinical animal models as well as patients with type 2 diabetes and complications such as diabetic nephropathy, diabetic retinopathy, diabetic neuropathy, cerebrovascular disease, coronary heart disease and peripheral artery disease compared to healthy controls. Severity of gut microbiota dysbiosis was associated with disease severity and restoration with probiotic administration in animal models and human patients has been associated with improvement of symptoms and disease progression. Characterizing the gut microbiota dysbiosis in different diseases and determining a causal relationship between the gut microbiota and disease can be beneficial in formulating therapeutic interventions for type 2 diabetes and associated complications. In this review, we present the most important findings regarding the role of the gut microbiota in type 2 diabetes and chronic complications as well as their underlying mechanisms.

## 1. Introduction

The gut microbiota is a complex ecosystem made up of a community of microorganisms that include trillions of bacteria spanning at least 1000 different species [1]. The gut microbiota is predominantly composed of bacteria but also contains other commensals such as archaea, viruses, fungi and protists [2]. All of these components are both relevant and important in understanding the relationship between the gut microbiota and the host.

Dysbiosis of the gut microbiota is primarily characterized by decreased diversity and abundance of bacteria and fungi, especially those associated with dysfunction and various pathologies [3]. Chief among them are cardiovascular, neuronal, immune and metabolic disorders [4] through the influence of bile acid metabolism, inflammatory status, insulin resistance and incretin secretion. This can lead to the emergence of obesity [5], metabolic syndrome and the onset of type 2 diabetes [6,7] through decreased glucose tolerance and insulin resistance [8]. The gut microbiota is an important player in chronic systemic inflammation secondary to endotoxemia caused by the release of endotoxins following bacterial death [9]. While the link between the gut microbiota and the onset and progression of diabetes is still under investigation, several studies to date have focused on the pathophysiology of diabetes, with few of them investigating the role of the gut microbiota in diabetes complications. This review summarizes the most important findings regarding the role of the gut microbiota in type 2 diabetes and describes its role on potential pathways that lead to chronic complications of diabetes. Modulation of the gut microbiota through the use of prebiotics, probiotics, synbiotics and fecal microbiota transplantation to restore metabolic deficits associated with these pathologies is also discussed.

## 2. Gut Microbiota, Type 2 Diabetes and Its Complications

Type 2 diabetes, like cardiovascular disease, cancer and chronic respiratory disease, is considered a chronic and noncommunicable disease responsible for 80% of premature deaths globally [10]. As of 2019, there were approximately 463 million cases of diabetes worldwide with an estimated 700 million by the year 2045 if current trends continue despite the variety of pharmacological interventions currently available [11]. 

Diabetes is characterized by high blood sugar levels that occur as a result of decreased pancreatic insulin production or decreased insulin sensitivity in tissues that typically respond to insulin signaling [12]. Poorly controlled diabetes and metabolic disorders associated with type 2 diabetes such as impaired lipid metabolism, the presence of oxidative stress and hypertension [13] can lead to both microvascular and macrovascular complications. Some microvascular complications of type 2 diabetes that involve small blood vessels include diabetic nephropathy, diabetic neuropathy and diabetic retinopathy. Conversely, common macrovascular complications that involve large blood vessels include cerebrovascular disease, coronary heart disease and peripheral vascular disease [14]. Other macrovascular complications of poorly controlled diabetes include congestive heart failure, impaired lipid metabolism, stroke, organ inflammation, weight gain, peripheral vascular disease and electrolyte imbalance [15].

Changes in interdependent metabolic pathways have also been observed in association with type 2 diabetes [16]. For example, coronary heart disease caused by impaired insulin metabolism can lead to dyslipidemia which is a risk factor for cardiovascular complications of diabetes [17]. Other specific factors known to contribute to the progression of diabetes complications include increased reactive oxygen species (ROS), chronic hyperglycemia and decreased antioxidant status [18]. The presence of these complications also leads to an overall decline in quality of life and an increase in mortality rate [19].

A plethora of studies have demonstrated a significant association between changes in the composition profile of gut microbiota and development of diabetes. In particular, perturbed *Bacteroidetes/Firmicutes* phylum eubiosis has been linked with increased intestinal permeability, with infiltration of bacteria byproducts through a leaky gut barrier triggering subsequent inflammatory responses characteristic of diabetes. On the other hand, several bacteria have been shown to exert a protective role by decreasing the risk of diabetes development through reduction in proinflammatory markers and maintaining intestinal barrier integrity. For example, *Lactobacillus fermentum, plantarum* and *casei*, *Roseburia intestinalis, Akkermansia muciniphila* and *Bacteroides fragilis* have all been shown to improve glucose metabolism and insulin sensitivity, and suppress proinflammatory cytokines. Notably, some drugs such as metformin which is commonly used for diabetes treatment have also been shown to alter the composition of the gut microbiota, suggesting that metformin interacts with the gut microbiota through modulation of inflammation, glucose homeostasis, gut permeability and short-chain fatty acid-producing bacteria [20]. Additionally, in patients with diabetes-associated gut dysbiosis, metformin promotes butyrate and propionate production, improving a patient’s ability to catabolize amino acids [21]. These changes coupled with increased levels of *Akkermansia* in the gut may contribute to the effects of metformin on glucose metabolism [22]. It appears that the metabolic factors associated with chronic low-grade inflammation and oxidative stress, which link gut microbiota dysbiosis and type 2 diabetes, are the same ones that influence the onset and progression of diabetic complications [23,24]. This relationship gives credence to the concept that modulation of the gut microbiota may be a promising strategy in the management of diabetes and associated complications as presented in the following sections.

### 2.1. Gut Microbiota in Diabetic Nephropathy

Diabetic nephropathy occurs in approximately 40% of patients with poorly managed diabetes [25], of which approximately 20% are hemodialysis patients [26], leading to end-stage renal disease, as well as cardiovascular complications [27]. Recent increases in the number of diabetic nephropathy and end-stage renal disease cases have been attributed to modern societal habits and lifestyle risks associated with diabetes and hypertension [28,29]. Similarly, increased stress on the kidneys due to hyperglycemia can lead to diabetic nephropathy as well as associated systemic inflammation, micro and macro albuminuria and proteinuria [30,31]. In addition, other factors such as genetics, age, obesity, high blood pressure and dyslipidemia [32,33] all have been shown to contribute to the progression of diabetic nephropathy. More recently, however, several studies have shown that dysbiosis of the gut microbiota can play a role in the development of chronic kidney disease [34]. In particular, the products of bacterial metabolism have been shown to influence the occurrence and progression of chronic kidney disease [35] while progression to renal failure lead to worsening of gut microbiota dysbiosis [36]. 

For example, the composition of the gut microbiota differs in both animals and people with chronic kidney disease. In studies using animals and humans with chronic kidney disease there was a decrease in the proportion of *Bifidobacterium* [37], *Bactemides* [38] and *Lactobacillus* [36]. Moreover, in patients with chronic kidney disease a decrease in the proportion of *Prevotella* [39,40], *Ruminococcaceae*, *Roseburia*, *Faecalibacterium* [40] and an increase in the proportion of *Parabacteroides* [39], *Enterococcus* [40], *Enterobacteriaceae* [36] and *Klebsiella* [40] have been reported. The increased proportions of *Bacterioidaceae* and *Clostridiaceae* in patients with chronic kidney disease have been associated with systemic inflammation [41]. In contrast, bacteria such as *Lactobacillaceae*, *Prevotellaceae* and *Bifidobacteriacea* [36], that are associated with anti-inflammatory effects and protecting intestinal barrier integrity were less represented in patients with chronic kidney disease [41]. In general, patients with chronic kidney disease show a decreased proportion of anaerobic bacteria [42]. Furthermore, bacterial DNA was present in the blood of 20% of patients with chronic end-stage renal disease who were not on dialysis. In these patients, the same bacterial genus was detected in their intestines along with increased biomarkers of low-grade inflammation [43]. 

Given that the imbalance of the gut microbiota influences many chronic diseases including type 2 diabetes and its complications, it follows that balancing the composition of the gut microbiota could be a strategy for controlling or even preventing disease. Some studies have analyzed the effects of probiotics [42,44] or synbiotics in modulating the gut microbiota in patients with chronic kidney disease [45,46]. In a clinical trial conducted in patients with stage 3 and stage 4 chronic kidney disease, urea nitrogen in the blood and uric acid concentration decreased after administering a mixture of *Lactobacillus acidophilus, Streptococcus thermophilus* and *Bifidobacterium longum* for six months [42]. In a similar study, the level of uric nitrogen in the blood also decreased after administration of dairy products containing *Lactobacillus* for two months [47]. When the effects of probiotics were tested in patients with chronic kidney disease on dialysis, administration of *Lactobacillus acidophilus* improved blood levels of dimethylamine and nitrodimethylamine [48], as well as lowered the level of dimethylamine and nitrosodimethylamine, a known carcinogen [49]. A summary of results of clinical trials examining the effects of probiotic intake in patients with type 2 diabetes and kidney disease is presented in Table 1. Thus, research has delineated specific changes in the gut microbiome associated with diabetic nephropathy as well as physiologic mechanisms underlying changes resulting from probiotic or symbiotic supplementation in patients with diabetic nephropathy. 

### 2.2. Gut Microbiota in Diabetic Retinopathy

In poorly controlled diabetes, the pressure inside the eye increases, and the accumulation of glucose in blood vessels can affect the health of the eye [15]. These processes are associated with microvascular complications in the eye including cataracts, glaucoma and retinopathy [56]. Diabetic retinopathy is a complication of poorly controlled diabetes that can result in blindness over time [57]. Increased activation of retinal microglia and infiltration of immune cells into the retina were found in diabetic retinopathy [58]. In addition, increased oxidative stress and inflammation can result in impaired functions of the renin-angiotensin system leading to metabolic disorders, including diabetic retinopathy [59,60,61,62,63,64,65]. Finally, gut microbiota dysbiosis have also been linked with development of diabetic retinopathy. 

The microbiota differs in composition throughout the body including the eye. For example, the internal eye compartment is sterile, however, the external compartment is exposed to environmental microorganisms [66]. While the overall gut microbiota is predominantly made up of *Firmicutes* and *Bacteroidetes* [67], the microbiota on the ocular surface is composed of primarily *Proteobacteria* and *Actinobacteria* [68,69]. In fact, *Proteobacteria, Actinobacteria* and *Firmicutes* have been shown to represent over 87% of all microorganisms present in the eye [70]. Several studies have reported an association between the imbalance of the gut microbiota or the microbiome on the ocular surface and various eye conditions. Furthermore, in humans, a significant decrease in the proportion of *Bacteroidetes* and *Actinobacteria* was observed in patients with diabetic retinopathy compared to healthy individuals. Additionally, significant increases in the proportion of *Acidaminococcus, Escherichia* and *Enterobacter* appear in the microbiota of patients with diabetic retinopathy compared to healthy controls [71]. Recent research showed a significant decrease in the *Mucoromycota* thread in patients with diabetic retinopathy compared to individuals without diabetic retinopathy. Likewise, in patients with type 2 diabetes and diabetic retinopathy, a decrease of 12 of the 18 genera present was observed [3]. Microbiota byproducts such as trimethylamine N-oxide (TMAO) derived from dietary choline metabolism have also been linked with diabetes retinopathy. For example, patients with diabetic retinopathy had higher plasma levels of TMAO and proinflammatory cytokines compared to diabetics without retinopathy [72], an effect associated with the severity of the disease. When microbiota composition was analyzed, there was a marked decrease in *Pasteurellaceae* in diabetic retinopathy [73]. Together, these findings support the concept that specific changes in the gut microbiome and mycobiome are associated with diabetic retinopathy.

Modulation of the gut microbiota profile via administration of probiotics has shown positive effects in preclinical models of diabetic retinopathy. For example, administration of recombinant *Lactobacillus paracasei* to mice with diabetic retinopathy reduced capillary cell loss and inflammatory expression of cytokines in the retina [74]. Similarly, administration of *Lactobacillus paracasei* secreting Ang- (1–7) to diabetic mice led to the amelioration of eye disease, by reducing retinal gliosis, inflammation and retinal capillary loss [75]. Lastly, modulation of the gut microbiota in mice with type 1 diabetes by administering *Lactobacillus rhamnosus* for four months resulted in weight loss, improved blood glucose and reduced intraocular pressure compared to the control group [76]. To date there are no studies investigating the effects of probiotic or symbiotic supplementation on diabetic retinopathy or the effects of modulating the microbiome on diabetic retinopathy in humans. 

### 2.3. Gut Microbiota in Diabetic Neuropathy

Chronic uncontrolled diabetes is associated with diabetic neuropathy, a neurodegenerative nutritional disease characterized by damage to peripheral nerves causing pain and numbness [56,77]. The characteristics of diabetic neuropathy are significant decline of peripheral innervations, increased neuronal inflammation, demyelination, axonal atrophy and the diminution of neuronal regenerative capacity [78]. Diabetic neuropathy is present in approximately 50% of diabetic patients [77] and affects many organs, resulting in various complications such as cardiovascular damage with symptoms of tachycardia, orthostatic hypotension, impaired intestinal transit, impaired gastric emptying, profuse sweating and hormonal imbalance. Diabetic peripheral neuropathy has been associated with certain factors, such as oxidative stress, activation of the polyol pathway and inflammation [79,80]. Insulin resistance is also implicated in the development of peripheral diabetic neuropathy. While peripheral diabetic neuropathy is a major complication of diabetes, its pathogenesis is not yet fully known.

Diabetic neuropathy has been linked to changes in the diversity of the gut microbiota and the increased presence of pathogens [81]. A comparison of the gut microbiota in patients with diabetic neuropathy, patients with diabetes without diabetic neuropathy and healthy individuals showed an increase in Firmicutes and Actinobacteria as well as a decrease in Bacteroidetes in patients with diabetic nephropathy when compared to patients with diabetes without diabetic neuropathy and healthy individuals. Furthermore, at the genus level, a decrease of Bacteroides and Faecalibacterium and an increase of Escherichia-Shigella, Lachnoclostridium, Blautia, Megasphaera and Rumincoccus torques were observed. It is hypothesized that these changes in the gut microbiota occur as a result of insulin resistance. In addition, elevated levels of Megasphaera have been directly correlated with Homeostatic Model Assessment for Insulin Resistance (HOMA-IR) scores in patients with diabetic neuropathy, which suggests that the presence of insulin resistance is associated with peripheral diabetic neuropathy [81]. 

Modulation of the gut microbiota by administration of *Bifidobacteria* and *Lactobacillus* or fecal transplantation can improve insulin resistance [82]. While several studies have characterized the gut microbiota in patients with diabetic neuropathy, the mechanisms by which gut microbiota acts on the onset and progression of diabetic neuropathy require further investigation. Recent research efforts have investigated the role of the gut microbiota in neurological disorders, including chronic pain [83]. Evidence shows that bacteria can directly activate nociceptors through constituent elements and byproducts [84,85]. For example, toxin produced by *Staphylococcus aureus*, called α-hemolysin, has been shown to induce spontaneous pain [86]. In patients with peripheral diabetic neuropathy, the presence of *Parabacteroidetes* is associated with amelioration of metabolic disorders and is positively correlated with CRP and Tauroursodeoxycholic acid (TUDCA) levels [81]. Additionally, the presence of *Parabacteroidetes* and changes in TUDCA levels may influence insulin resistance and the onset of dyslipidemia, which in turn affect the onset of peripheral diabetic neuropathy [81].

It is known that modulation of the gut microbiota can influence the central and peripheral nervous system, in a bidirectional matter through gut-microbiota-brain axis [83]. There are currently no pharmacological interventions available to treat diabetic neuropathy and the associated decline in quality of life that it may cause. Because of this, further research is required to investigate the effects of taking probiotic or synbiotic dietary supplements to prevent, control or even treat diabetic neuropathy. 

### 2.4. Gut Microbiota in Cerebrovascular Disease

Stroke is a major cause of disability worldwide and diabetes is one of many factors that increase stroke risk [87]. Additionally, poor blood sugar management negatively influences progression of cerebrovascular disease and increases mortality [88]. In most cases, it is difficult to determine with certainty what caused a stroke; however, recently research showed a link between gut microbiota dysbiosis and stroke incidence [89]. This may be due to the ability of the gut microbiota to interact with the central nervous system through endocrine, neuronal and immune pathways, directly affecting brain chemistry [90]. 

The composition of the gut microbiota changes in both rodents and humans after the onset of acute ischemic stroke. In a preclinical study using a rodent stroke model, increased amounts of *Akkermensia municiphila* and *Clostridia spp.* were noted in the experimental group post-stroke compared to the control animals [91]. Similarly, in human stroke patients, an increase of *Lactobacillus ruminis* and a decrease in *Lactobacillus sakei* was observed compared to the control group. [92,93]. Additionally, the gut microbiota of stroke patients included several species that produce short-chain fatty acids, such as *Odoribacter, Akkermensia, Ruminococcaceae UCG-005* and *Victivallis* [93]. Dysbiosis of the gut microbiota that develops post-stroke leads to impairment of neuroinflammatory processes that affect stroke progression.

Symptomatic atherosclerosis has been associated with dysbiosis of the gut microbiota as well, supporting a potential link between the gut microbiota, cardiovascular and cerebrovascular diseases [94]. One study that investigated a group at risk of developing a stroke in China showed changes in gut microbiota composition such that there was an increase in the amount of opportunistic pathogenic bacteria, including *Enterobacteriaceae* and *Veillonellaceae*, as well as lactate-producing bacteria including *Bifidobacterium* and *Lactobacillus*. Furthermore, there was a reduction in butyrate-producing bacteria, including *Lachnospiraceae* and *Ruminococcaceae*, in people at high stroke risk compared to low-risk individuals. Based on these data, it is possible that dysbiosis of the gut microbiota alone may represent a stroke risk factor [95].

Trimethylamine-N-oxide (TMAO) is a commonly studied metabolite when considering the link between the gut microbiota and stroke risk. This metabolite is the result of the transformation of phosphatidylcholine and l-carnitine into trimethylamine, which is then absorbed and oxidized by hepatic flavin monooxygenase to form TMAO [96]. While some studies have shown an association between TMAO, atherosclerosis and the risk of stroke, the mechanisms by which this association occurs are not well understood. There is a significant correlation between TMAO levels and the amount of pro-inflammatory intermediate monocytes observed; therefore TMAO is believed to influence inflammation by promoting the growth of proinflammatory monocytes [97]. Other proposed mechanisms for the formation of TMAO associated with stroke or cerebrovascular accident include the promotion of platelet hyperreactivity [98], irregular cholesterol metabolism [99] and promotion of foam cell formation [100]. TMAO is also associated with other ischemic stroke risk factors such as arterial fibrillation [101] and diabetes [102]. 

Several studies have shown an association between gut microbiota dysbiosis and atherosclerosis in patients on a phosphatidylcholine-rich diet [100]. Increased TMAO levels were also associated with an increased risk of cardiovascular disease [100]. A study of Chinese patients with high blood pressure showed that increased TMAO levels were associated with increased stroke risk as well [103]. Jia Yin et al. observed that the level of TMAO in patients with a history of stroke or transient ischemic attack (TIA) was significantly lower than in the control group of asymptomatic individuals. Furthermore, patients with stroke and TIA also had a different gut microbiota composition than those in the control group. The gut microbiota of patients that had suffered from a stroke or TIA was characterized by an increase in the amount of harmful pathogenic bacteria *Enterobacter, Megasphaera, Oscillibacter* and *Desulfovibrio*, and a decrease in the amount of beneficial or commensal bacteria, such as *Bacteroides, Prevotella* and *Faecalibacterium*. Moreover, this study emphasized the association between dysbiosis of the gut microbiota and the severity of cerebrovascular disease [104]. Based on these data, evaluating the gut microbiota could be an invaluable metric when assessing stroke risk in patients. 

Preclinical studies investigating the use of probiotic supplementation to improve gut dysbiosis associated with cerebrovascular disease show promising results. In mice, administration of a 10^7^ CFU / mL mixture of *Bifidobacterium breve, Lactobacillus casei, Lactobacillus bulgaricus* and *Lactobacillus acidophilus* 14 days prior to an ischemic event significantly reduced the size of the stroke by 52%. Furthermore, this administration of probiotics led to a significant decrease in the content of malondialdehyde and TNF-α in the ischemic tissue of the brain. Despite the observed reduction of stroke size, the administered probiotics did not improve the neurological function of the experimental group mice compared to the control group [105]. While the preclinical data are promising, further clinical research is needed to investigate the effect of probiotic supplementation on human gut dysbiosis and associated cerebrovascular disease. 

### 2.5. Gut Microbiota in Coronary Heart Disease

Coronary artery disease is the leading cause of morbidity and mortality worldwide, as well as an important determinant of long-term prognosis in patients with diabetes. Diabetic patients with heart disease have a two-to-four times higher risk of mortality [106]. It is known that the gut microbiota plays a critical role in essential metabolic processes, such as cholesterol and uric acid metabolism in addition to influencing processes such as oxidative stress and inflammatory reactions through metabolites, which can lead to atherosclerosis or coronary heart disease [107]. Because hypercholesterolemia is a known risk factor for coronary artery disease, and dysbiosis of the gut microbiota can affect cholesterol metabolism, it follows that dysbiosis of the gut microbiota can be a risk factor for coronary artery disease [108]. Gut microbiota dysbiosis also affects the development of hypercholesterolemia by influencing the metabolism of cholesterol in the liver and by altering bile acids, which in turn affect circulating cholesterol levels [109]. Recently, a growing number of both preclinical and clinical studies have implicated gut microbiota in the occurrence of coronary heart disease. For example, patients with coronary artery disease showed increases in *Collinsella* bacteria [94], mature lactobacilli [110], *Escherichia-Shigella* [111], *Enterococcus* [111] and the ratio of *Firmicutes* to *Bacteroides* [112]. Conversely, significant decreases in *Roseburia* and *Eubacterium spp.* [94], *Bacteroides* (*Bifidobacterium* and *Prevotella*) [110] and butyrate-carrying bacteria, such as *Faecalibacterium*, *Roseburia* and *Eubacterium rectalae* were observed in the gut microbiota of patients with coronary artery disease compared to healthy individuals [111].

In mice, antibiotic-induced changes in the gut microbiota significantly altered host metabolism and determined the severity of subsequent myocardial infarction [113]. On the other hand, addition of *Lactobacillus plantarum* and *Lactobacillus rhamnosus* reduced the size of the infraction, ameliorated left ventricular hypertrophy and improved left ventricular function post- infarction [114]. In humans, dysbiosis of the gut microbiota can lead to coronary artery disease, hypertension and heart failure [100]. For example, one study showed a higher frequency of coronary artery disease in the presence of a low proportion of intestinal bacteria [115]. It has been suggested that the gut microbiota influences the development of coronary artery disease by producing metabolites such as bile acids, coprostanol, short-chain fatty acids and TMAO. TMAO levels are strongly associated with coronary artery disease risk. Uric acid serum levels could also be an independent risk factor for coronary artery disease. Furthermore, elevated uric acid levels in patients with coronary artery disease are linked to dysfunction of the gut microbiota [116]. Patients with coronary artery disease showed a reduction in primary plasma bile acids and an increased ratio of secondary to primary bile acids in patients with heart failure [117], which could affect disease progression. 

Studies investigating the effects of probiotic supplementation on the gut microbiota, diabetes and coronary artery disease have shown promising results. In patients with coronary artery disease, probiotics reduced blood lipids, thus reducing the risk of coronary artery disease [118]. Additionally, a group of 20 men with coronary artery disease who received a probiotic drink containing *Lactobacillus plantarum* 299 for six weeks showed improvement of endothelial vascular function and decreased systemic inflammation [119]. Another study monitored the effects of taking a probiotic supplement containing *Bifidobacterium bifidum* 2 × 10^9^, *Lactobacillus casei* 2 × 10^9^, *Lactobacillus acidophilus* 2 × 10^9^ CFU/day in patients with diabetes and coronary heart disease. After 12 weeks of this protocol, patients exhibited improved glycemic control, increased HDL-cholesterol, low total cholesterol to HDL-cholesterol ratio and a reduction in oxidative stress biomarkers [120]. In short, the use of probiotics is a promising approach to treatment of individuals with diabetes-associated gut dysbiosis and coronary artery disease. 

### 2.6. Gut Microbiota in Peripheral Vascular Disease

Peripheral arterial disease (PAD) is a severe complication of late-stage type 2 diabetes. PAD is often associated with critical limb ischemia and gangrene. Diabetic foot is one example of this that often occurs with poorly controlled diabetes. This is characterized by hyperglycemia, hyperinsulinemia and dyslipidemia [121] and can result in increased systemic inflammation and oxidative stress as well as diabetic foot ulceration [122]. More than 25% of patients with diabetes are at risk of developing diabetic foot and associated ischemia, neuropathy or infection [123,124]. Lesions such as ulcers that develop in diabetic patients experience difficulty healing due to decreased blood flow caused by the accumulation of lipid plaques on the walls of the vessels. This delay in healing can cause inflammation and gangrene [125]. In addition, poor perception of pain caused by associated diabetic neuropathy often leads to delays in identifying and diagnosing diabetic peripheral vascular disease [126] and high limb amputation rate [127]. Despite wide prevalence and the severity of its consequences, peripheral vascular disease is the least studied vascular complication of diabetes [128]. 

The effects of probiotic supplementation on peripheral vascular disease and lesion healing have been investigated in preclinical rodent models. When kefir was administered to rats, it improved lesion healing due to the lactic acid producing bacteria that inhibits proliferation of pathogenic microbes. Other components of kefir, such as polysaccharides improved wound healing by stimulating the innate immune response against pathogens present in the wound [129]. The effects of probiotic supplementation on diabetic wound healing associated with peripheral vascular disease have also been investigated in humans. Diabetic foot patients who received a probiotic protocol for 12 weeks showed a reduction in the length, width and thickness of the diabetic foot ulcer. Furthermore, the probiotic supplement administered, consisting of *Lactobacillus acidophilus*, *Lactobacillus casei*, *Lactobacillus fermentum* and *Bifidobacterium bifidum* (2 × 10^9^ CFU/g each), led to improvements in plasma glucose, serum insulin and the QUICKI indicator [130]. While not thoroughly investigated, it has been suggested that the mechanism by which probiotics improve diabetic foot ulcers is similar to the one involved in improving lesions in other areas of the body, by modulating the local immune response [131]. Thus, increasing the diversity and richness of the gut microbiota, and establish eubiosis through probiotic supplementation may provide some benefits to patients with complications of diabetic peripheral vascular disease by improving glycemic control, insulin, lipid metabolism and incretins [132] (Table 2 and Figure 1). For example, in a proof-of-concept, randomized double-blind controlled clinical trial study, Depommier et al. showed that supplementation for three months with *A. muciniphila* significantly improved insulin sensitivity, reduced insulinemia, plasma total cholesterol and inflammation [133]. These results show that intervention with specific bacteria strains may prove a useful strategy in improving metabolic parameters associated with diabetes and its complications. Indeed, several bacteria with enhanced functional characteristics in treating specific host diseases have been defined as next generation probiotics (NGP). Among them, *Akkermansia muciniphila, Ruminococcus bromii, Faecalibacterium prausnitzii, Anaerobutyricum hallii* and *Roseburia intestinalis* have gained considerable interest and have been the primary candidates. In particular, *A. muciniphila* have been associated with improved metabolic endotoxemia, amelioration of metabolic syndrome phenotype, improved lipid and glucose metabolism and may serve as diagnostic tool for dietary interventions. Likewise, *Faecalibacterium prausnitzii* has been shown to exert anti-inflammatory action and has been proposed as a biomarker for the development of gut diseases and for assessing dietary interventions in intestinal inflammatory conditions [134] (Table 2). Based on these findings, several novel food and pharma supplements have been developed with profound beneficial effects in protecting from specific metabolic disorders and other metabolic risks.

## 3. Conclusions and Future Perspectives

For the past decade or so, owing to rapid methodological advances in genome sequencing of microbes, an avalanche of studies has rushed to uncover the potential contribution of the so called “forgotten organ” (i.e., gut microbiota) in multiple pathologies, including metabolic disorders. While significant strides have been made toward understanding the complex interaction between bacteria and the host, particularly at the biochemical, cellular and molecular level, we are still in the early stages when it comes to our understanding of whether gut bacteria play a direct role in prevention, development and treatment of diseases. As it is the case with most pathologies in which the effects of gut microbiota have been studied, the development of diabetes and its complications have been linked with the state of dysbiosis of the gut microbiota. This, in and of itself, raises a wide range of questions, since “dysbiosis” is a loose term used to characterize a disequilibrium, in a given organism and time [177]. As noted throughout this review, it is well documented that diabetes and its complications are characterized by systemic inflammation, therefore it is not surprising that numerous studies focused on examining the anti-inflammatory effects of certain bacteria such as *Roseburia* in patients with coronary artery disease, *Lachnospiraceae* in patients at high stroke risk and *Faecalibacterium* in patients with diabetic nephropathy, diabetic neuropathy, cerebrovascular disease or coronary artery disease. As such, low abundance of anti-inflammatory bacteria, along with the increased abundance of pro-inflammatory bacteria has been attributed to the onset and progression of complications of diabetes. Similarly, bacterial metabolites such as SCFA and TMAO have been shown to influence host physiology and improve disease outcome. Notwithstanding such promising findings, we are still very much grasping with the demonstration, beyond doubt, of a causal relationship between gut bacteria and diabetes and its complications. Whereas preclinical studies are promising and show direct effect of some bacteria on certain metabolic and clinical parameters of diabetes, the results in humans are less promising, with few clinical trials and by and large, have been inconsistent. Thus, for the modulation of gut microbiota via prebiotics, probiotics, FMT or other means to be part of any therapeutic protocol in diabetes and its complications, its causal effect in these diseases must be defined and clinically demonstrated. Preclinical animal models such as germ free or antibiotic treated animals have been useful in examining host-microbiota interactions via controlling the effects of individual bacteria, through monocolonization or combined bacteria therapy, however, they each come with significant caveats that often preclude generalization of findings to human disease prevention and treatment. Considering that bacterial strains of the same species may differ in up to 30% of their genomic structure when compared by taxonomic analysis, it follows that gut microbiota must be viewed and analyzed as a system. Similarly, microbial metabolites associated with the gut microbiota, type 2 diabetes and associated complications that act synergistically must be analyzed and their effects tested [178]. It is equally important to examine the dynamical changes in the composition profile and production of metabolic byproducts of gut microbiota prior, during and after the onset of diabetes and its complications in order to determine dynamic changes during disease progression. While to date the list of bacteria reported to affect several parameters characteristic of diabetes complications is steadily increasing, very few have been studied as therapeutic approaches in these pathologies. Likewise, efforts should be dedicated toward identification of bacteria signatures and metabolites that will allow early detection of disease risks, and the mechanisms involved, making possible to personalize therapeutic intervention based on individual’s needs, stage and particularities of the disease. Therefore, modulation of the gut microbiota through prebiotics, probiotics, synbiotics or fecal microbiota transfer may have beneficial effects in the management of diabetes and associated complications; however, further research involving human trials should be high on the list.

## Figures and Tables

**Figure 1 nutrients-14-00166-f001:**
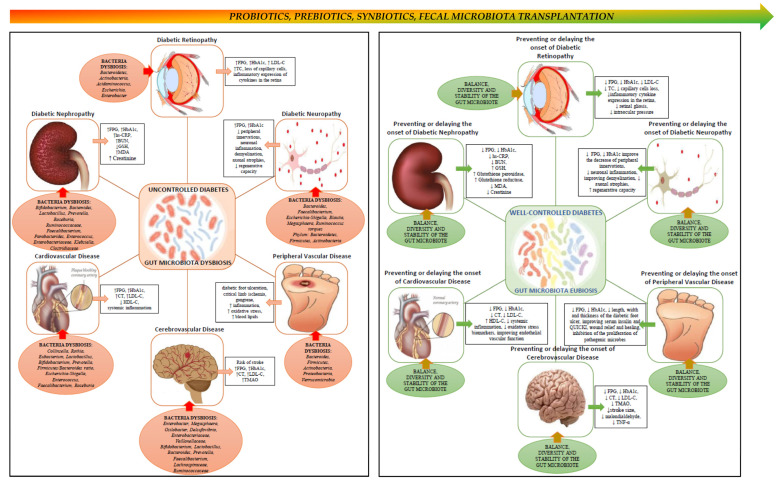
Schematic view of the link between gut microbiota, diabetes and chronic complications of diabetes. The **left** side panel depicts chronic micro- and macrovascular complications of diabetes, and associated changes in the composition of the gut microbiota. Poorly controlled diabetes leads to chronic complications over time, and dysbiosis of the gut microbiota seems to promote the onset and progression of these complications. The **right** panel depicts the potential effects of restoring gut microbiota eubiosis in ameliorating, preventing or delaying the onset of chronic complications of diabetes, via probiotics, prebiotics, symbiotics or by fecal microbiota transplantation. ↑, increase; ↓, decrease.

**Table 1 nutrients-14-00166-t001:** Effects of probiotics on type 2 diabetes and kidney disease.

Reference	Design	Probiotic Source	Probiotic Dose, CFU	Study Period (wk/d)	Effects
[50]	RD, DB, CT	tablet	*L. acidophilus strain ZT-L1*, *B. bifidum strain ZT-B1*, *L. reuteri strain ZT-Lre,* *L. fermentum strain ZT-L3* 8 × 10^9^ CFU/d	12 wk	S↓ FG, I, HOMA-IR, TG, VLDL, TC/HDL-C ratio, hs-CRP, MDA, AGEs, BUN, creatinine, urine protein S↑ QUICKI, HDL-C, GSH, CG = HbA1c, LDL-C, NO, TAC
[51]	RD, DB, CT	soy milk	*L. plantarum A7*	8 wk	S↓ albuminuria, serum creatinine, serum interleukin-18, serum sialic acid S improvment in estimated GFR
[52]	RD, DB, CT	capsule	*L. acidophilus* *L. casei* *B. bifium*	12 wk	S↓ FG, I, HOMA-IR, HbA1c, hs-CRP, MDA, SGA score, TIBC S↑ QUICKI =HOMA-B, TG, VLDL, CT, LDL-C, HDL-C, NO, TAC, GSH, GFR, creatinine, BUN, albumin, Na, K
[53]	RD, DB, CT	honey	*Bacillus coagulans T4* (IBRC-N10791) 10^8^ CFU/g	12 wk	S↓ I, HOMA-IR, CT/HDL-C ratio, hs-CRP hs-CRP, MDA, creatinine S↑ QUICKI =FG, TG, VLDL, CT, LDL-C, HDL-C, NO, TAC, GSH, BUN
[54]	RD, DB, CT	soy milk	*L. plantarum A7* 2 × 10^7^ CFU/mL	8 wk	S↓ Cys-C, PGRN, NGAL =sTNFR1
[55]	RD, DB, CT	soy milk	*L plantarum A7 (KC 355240, LA7)* 2 × 10^7^ CFUmL	8 wk	S↑ Glutathione, Glutathione peroxidase, Glutathione reductase S↓ Oxidized glutathione =MDA, 8-iso-PGF2a, TAC

RD, randomized; DB, double-blind; CT, clinical trial; T2DM, type 2 diabetes mellitus; L., Lactobacillus, B., Bifidobacterium; CFU, colony-forming units; wk, weeks; d, days; FG, fasting glucose fasting blood glucose, fasting plasma glucose, glycemia, fasting blood sugar; HbA1c, hemoglobin A1c; I, serum insulin concentration, insulin concentration, serum insulin level, insulin; HOMA-IR, homeostasis model of assessment-estimated insulin resistance; QUICKI, quantitative insulin sensitivity check index; TG, triglycerides; VLDL, very-low-density lipoprotein; TC, total cholesterol; HDL-C, high-density lipoprotein; LDL-C, low-density lipoprotein; hs-CRP, high-sensitivity C-reactive protein; NO, nitric oxide; TAC, total antioxidant capacity; GSH, total glutathione; MDA, malondialdehyde; AGEs, advanced glycation end products; BUN, blood urea nitrogen; CG, Cockcroft–Gault formula to estimate creatinine clearance; HOMA-B, homeostasis model of assessment–estimated b-cell function; GFR, glomerular filtration rate; SGA, subjective global assessment; TIBC, total iron binding capacity; Na, sodium; K, potassium; Cys-C, cystatin C; PGRN, Progranulin; NGAL, neutrophil gelatinase-associated lipocalin; sTNFR1, soluble tumor necrosis factor receptor 1; 8-iso-PGF2a, 8-iso-prostaglandin F2 alpha; =, non significan; S, significant; ↑, increase; ↓, decrease.

**Table 2 nutrients-14-00166-t002:** Effects of probiotic or synbiotic on glycemia, insulin, lipid metabolism and incretins.

Reference	Year	Location	Design	Participants, Age, Nr. Treated/ Nr. Controls	Probiotic Source	Probiotic Dose, CFU	Study Period (wk/d)	Glycemia	Insulin	Lipid Metabolism	Incretins
[135]	2002	Poland	RD, DB, CT	Healthy participants 35–45 y 18/18	rose-hip drink	*L. plantarum 299v,* 5 × 10^7^ CFU/mL	6 wk	=FG	=I	=TC, LDL-C, HDL-C, TG, lipoprotein(a)	S↓ leptin
[136]	2006	Australia	DB, PC, parallel design trial, single centre	Healthy volunteers 30–75 y 23/21	capsule	*L. fermentum,* 2 × 10^9^ CFU	10 wk	=FG	-	=LDL-C, TC, HDL-C, TGL	-
[137]	2009	Finland	RD, prospective, parallel-group	Pregnant women 29.7/30.1/30.2 y 85/86/85	capsule	*L. rhamnosus GG, ATCC 53 103,* *B. lactis Bb12,* 10^10^ CFU/d each	4 wk	S↓ FG, =HbA1c	S↓ I, HOMA, S↑ QUICKI	-	-
[138]	2010	Denmark	RD, PC, DB	T2DM/non-diabetic 48–66 y 24/24	capsule	*L. acidophilus NCFM,* 1 g; about 10^10^ CFU	4 wk	-	=QUICKI	-	-
[139]	2012	Iran	DB, RD, CT	T2DM 30–60 y 32/32	yogurt	*L. acidophilus La5,* 7.23 × 10^6^–1.85 × 10^6^ CFU/g *B. lactis Bb12*, 6.04 × 10^6^ CFU/g–1.79 × 10^6^ CFU/g	6 wk	S↓ FG, HbA1c	=I	-	-
[140]	2012	Brazil	DB, PC, RD	Healthy participants 50–65 y 10/10	shake	*L. acidophillus*, 4 × 10^8^ CFU/100 mL *B. bifidum* 4 ×10^8^ CFU/100 mL 1 g/100 mL FOS	30 d	S↓ FG	-	S↑ HDL-C =TC, TG	-
[141]	2012	Canada	DB, PC, multi-center study	Healthy hypercholester-olemic human subjects 20–75 y 67/64	capsule	*L. reuteri NCIMB 30242,* 2.9 × 10⁹ CFU	9 wk	=FG	-	-	-
[142]	2012	Denmark	DB, PC, RD	Ob adolescents 12–15 y 27/23	capsule	*L. salivarius Ls-33 ATCC SD5208,* 10^10^ CFU	12 wk	=FG	=I, HOMA-IR	=TC, HDL-C, LDL-C, TG	-
[143]	2013	Iran	RD, DB, PC, CT	T2DM 35–70 y 27/27	capsule	*L. acidophilus*, 2 × 10^9^ CFU *L. casei*, 7 × 10^9^ CFU *L. rhamnosus*, 1.5 × 10^9^ CFU *L. bulgaricus*, 2 × 10^8^ CFU *B. breve*, 2 × 10^10^ CFU *B. longum*, 7 × 10^9^ CFU *S. thermophiles*, 1.5 ×10^9^ CFU 100 mg FOS	8 wk	S↓ FG	S↑ I, HOMA-IR	S↑ LDL-C	-
[144]	2013	Iran	RD, DB, CT	Patients with NASH 18–75 y 34/36	tablet	*L. acidophilus,* 1 × 10^8^ CFU *L. casei,* 5 × 10^8^ CFU *L. rhamnosus,* 7.5 × 10^7^ CFU *L. bulgaricus,* 1.5 × 10^8^ CFU *B. breve,* 5 × 10^7^ CFU *B. longum,* 2.5 × 10^7^ CFU *S. thermophilus,* 5 × 10^7^ CFU 350 mg FOS	24 wk	S↓ FG	-	S↓ TC, TG	-
[145]	2013	Korea	single center, RD, DB, PC, CT	Ob volunteers 19–60 y 31/31	capsule	*L. gasseri* BNR17, 10^10^ CFU 25% FOS	12 wk	=FG, HbA1c	=I	=TC, TG, LDL-C, HDL-C,	-
[146]	2013	Russian Federation	RD, DB, PC, parallel pilot study	Patients with metabolic syndrome 30–69 y 25/15	cheese	*L. plantarum* TENSIA, 1.5 × 10^11^ CFU/g	3 wk	=FG	-	=TC, LDL-C, HDL-C, TG	-
[147]	2013	Iran	RD, SB, CT	Pregnant women 37/33 18–30 y	yogurt	*L. acidophilus LA5,* *B. animalis BB12,* 1 × 10⁷ CFU	9 wk	=FG	S↓ I, HOMA	-	-
[148]	2014	Iran	RD, DB, cross-over CT	T2DM 35–70 y 62/62	package	*L. sporogenes,* 27 × 10^7^ CFU 1.08 g inulin	6 wk	=FG	S↓ I =HOMA-IR	=CT, LDL-C, TG, HDL-C	-
[149]	2014	Iran	RD, DB, CT	T2DM ov/ob obese 53.00 ± 5.9/ 49.00 ± 7.08 y 22/22	yogurt	*B. lactis* Bb12, *L. acidophilus* strain La5, 3.7 × 10^6^ CFU/g	8 wk	S↓ HbA1c =FG	-	-	-
[150]	2014	Ireland	PC, DB, RD	Ob pregnant women, 31.4 ± 5.0/31.0 ± 5.2 y 63/75	capsule	*L. salivarius UCC118,* 10^9^ CFU	4 wk	=FG	=I, HOMA-IR	=TC, HDL-C, LDL-C, TG	-
[151]	2014	Australia	RD, DB, parallel study	Ov >55 y 40/37/39/40	Yogurt/ capsule	*L. acidophilus La5*, *B. lactis Bb12,* 3 × 10^9^ CFU/d	6 wk	S↑ FG =HbA1c	S↑ HOMA-IR =I	-	-
[152]	2014	India	RD, CT, DB	Ov/ob healthy adults 40–60 y 15/15/15/15	capsule	*B. longum*, *B. infantis*, *B. breve*, *L. acidophilus*, *L. paracasei*, *L. bulgaricus*, *L. plantarum,* *S. thermophilus*. 112.5 × 10^9^ CFU/capsule	6 wk	S↓ FG	S↓ I, HOMA-IR	S↓ TC, TG, LDL-C, VLDL-C S↑ HDL-C	-
[153]	2014	Japan	SB, PC, within-subject, repeated-measure intervention trial	Adults with hypertriacylglycerolemia, 51.1 ± 6.6 y 10/10	fermented mil	*L. gasseri* SBT2055 (LG2055), 5 × 10^10^ CFU/100 g	4 wk	S↑ HbA1c =FG	=I	S↓ NEFA =TG, Apo B-48, TC, LDL-C, HDL-C	-
[154]	2014	Iran	RD, DB, PC, CT	NAFLD >18 y 26/26	capsule	*L. casei*, *L. rhamnosus*, *S. thermophilus*, *B. breve*, *L. acidophilus*, *B. longum*, *L. bulgaricus* 2 × 10^8^ CFU 250 mg FOS	28 wk	S↓ FG	S↓ I, HOMA-IR	-	-
[155]	2014	Iran	RD, DB, PC pilot study	Patients with MS >18 y 19/19	capsule	*L. casei*, *L. rhamnosus*, *S. thermophilus*, *B. breve*, *L. acidophilus*, *B. longum*, *L. bulgaricus* 2 × 10^8^ CFU 250 mg FOS	28 wk	S↓ FG	S↓ I, HOMA-IR S↑ QUICKI	=LDL-C S↓ TG, CT S↑ HDL	-
[156]	2014	Iran	RD, PC, CT	Pregnant women 18–35 y 26/26	food	*L. sporogenes,* 1 × 10^7^ CFU 0.04 g inulin	9 wk	=FG	S↓ I, HOMA-IR, HOMA-B S↑ QUICKI	-	-
[157]	2014	Iran	RD, DB, CT	T2DM 35–70 y 26/26/26	bread	*L. sporogenes,* 1 × 10^8^ CFU 0.07 g inulin	8 wk	=FG	-	S↓ TG, VLDL-C, TC/HDL-C S↑ HDL-C = TC, LDL-C, HDL-C	-
[158]	2015	Germany	DB, RD, prospective, longitudinal pilot	Lean/ob participants 40–65 y 11/10	capsule	*L. reuteri,* 2 × 10^10^ CFU	8 wk	=blood glucose levels during OGTT	S↑ QUICKI in lean participants compared with obese	-	S↑ GLP-1, GLP-2
[159]	2015	Iran	RD, DB, PC, CT	T2DM 35–65 y 30/30	fermented milk (kefir)	*L. acidophilus,* 3 × 10^6^–25 × 10^6^ *L. casei,* 2 × 10^6^–15 × 10^6^ *B. lactis,* 0.5 × 10^6^–8 × 10^6^	8 wk	S↓ HbA1c, FG	-	=TG, TC, LDL-C, HDL-C	-
[160]	2015	India	RD, CP, SB, pilot study	Healthy participants 20–25 y 15/15/15	capsule	*L. salivarius UBL S22,* 2 × 10^9^ CFU 10 g/d FOS	6 wk	S↓ FG	S↓ I, HOMA-IR	S↓ TG, CT, LDL-C S↑ HDL-C	-
[161]	2015	Denmark	CT, DB, RD, PC, two-arm parallel	Young healthy adults 20–45 y 32/32	capsule	*L. casei W8,* 10^10^ CFU	4 wk	=FG	=I	S↓ TG =CT, HDL-C, LDL-C	=GLP1
[162]	2016	Iran	RD, DB, PC, CT	GDM, 18–40 y 30/30	capsule	*L. acidophilus*, 2 × 10^9^ CFU/g *L. casei,* 2 × 10^9^ CFU/g *B. bifidum*, 2 × 10^9^ CFU/g	6 wk	S↓ FG	S↓ I, HOMA-IR, HOMA-B S↑ QUICKI	S↓ TG, VLD-C, =TC, HDL-C	-
[163]	2016	Iran	RD, SB, CT	Ob/ov subjects 18–50 y 44/45	yogurt	*L. acidophilus LA5*, *B. lactis BB12* 1 × 10^7^ CFU	12 wk	S↓ 2-h postprandial glucose, HbA1c =FG	S↓ HOMA-IR, I	S↓ TC, LDL-C =HDL-C, TG	-
[164]	2016	Estonia	preliminary, open label study	Clinically healthy volunteers 50–75 y	capsule	*L. fermentum ME-3 (LFME-3),* 6 × 10^9^ CFU	4 wk	S↓ HbA1c	S↓ HOMA-IR	S↓ LDL-C, oxLDL, TC, TG, TG/HDL-C ratio S↑ HDL-C	-
[165]	2017	Sweden	RD, PC	T2DM 50–75 y 15/15/16	stick pack	*L. reuteri DSM 17938,* 10^8^ CFU/day *L. reuteri DSM 17938,* 10^10^ CFU/day	12 wk	=FG =HbA1c	S↑ QUICKI	=CT, HDL, LDL, TGL	-
[17]	2017	Iran	RD, CT	T2DM, ov, CHD patients 40–85 y 30/30	capsule	*L. acidophilus,* 2 × 10^9^ *L. casei,* 2 × 10^9^, *B. bifidum,* 2 × 10^9^ CFU/g 800 mg inulin	12 wk	S↓ FG	S↓ I, HOMA-B S↑ QUICKI =HOMA-IR	S↑ HDL-C =TG, TC, LDL-C, VLDL-C, TC/HDL-C ratio	-
[166]	2017	Malaysia	RD, DB, parallel-group, CT	T2DM, 30–70 y 68/68	sachet	*L. acidophilus*, *L. casei*, *L. lactis*, *B. bifidum*, *B. longum*, *B. infantis,* 10^10^ CFU/d each	12 wk	S↓ HbA1c =FG	S↓ I =HOMA-IR, QUICKI	=TC, TG, LDL-C, HDL-C	-
[167]	2017	Brazil	DB, RD, PC, CT	T2DM 35–60 y 25/25	fermented goat milk	*L. acidophilus La-5*, *B. lactis BB-12,* 10^9^ CFU/d each	6 wk	S↓ FS =HbA1c, FG	= I, HOMA-IR	S↓ TC, LDL-C =HDL, VLDL, TG. CT/HLD-C ratio	-
[9]	2017	Saudi Arabia	DB, RD, CT	T2DM 30–60 y 48/46	sachet	*B. bifidum* W23, *B. lactis* W52, *L. acidophilus* W37, *L. brevis* W63, *L. casei* W56, *L. salivarius* W24, *Lactococcus lactis* W19, *Lactococcus lactis* W58, 2.5 × 10^9^ CFU/g	12 wk	=FG	S↓ HOMA-IR = I	=TG, TC, HDL-C, LDL-C, TC/HDL ratio	-
[168]	2017	Iran	RD, DB, PC, CT	NAFLD patients with normal or low BMI >18 y 25/25	capsule	*L. casei*, *L. rhamnosus*, *S. thermophilus*, *B. breve*, *L. acidophilus*, *B. longum*, *L. bulgaricus* 2 × 10^8^ CFU	28 wk	S↓ FG	=HOMA-IR, I, QUICKI	=LDL-C, HDL-C, TC S↓ TG	-
[169]	2018	Taiwan	DB, RD, PC	T2DM 25–70 y 25/25/24	capsule	*ADR-1* (live *L. reuteri*), 4 × 10^9^ CFU cells of ADR-3 (heat-killed *L. reuteri*), 2 × 10^10^ CFU	24 wk	=fasting blood glucose S↓ HbA1c in liver ADR-1 =HbA1c in heat-killed ADR03,	=I, HOMA-IR	=LDL-C, free fatty acids S↓ TC in ADR-1	-
[170]	2018	Iran	DB, RD, PC, parallel-group, CT	Prediabetes 40/40/40 35–75 y	powder	*L. acidophilus*, *B. lactis*, *B. bifidum*, *B. longum* 1 × 10^9^ CFU/each inulin	24 wk	S↓ FG, HbA1c	S↓ I, HOMA-IR S↑ QUICKI =HOMA-B	-	-
[171]	2018	Ukraine	DB, single center RD, CT	T2DM, ov 18–75 y 31/22	sachet	14 alive probiotic strains of *L.+ Lactococcus,* 6 × 10^10^ CFU/g *B.,* 1 × 10^10^ CFU/g, *Propionibacterium,* 3 × 10^10^ CFU/g, *Acetobacter,* 1 × 10^6^ CFU/g	8 wk	S↓ HbA1c =FG	S↓ HOMA-IR =I	-	-
[172]	2018	Iran	RD, DB, PC, CT	T2DM, CHD 45–85 y 30/30	capsule	*L. acidophilus*, *B. bifidum*, *L. reuteri*, *L. fermentum* 8 × 10^9^ CFU/g	12 wk	=FG	S↓ I, HOMA-IR S↑ QUICKI	S↑ HDL-C =LDL, TC, TG, VLDL-C	-
[82]	2018	Saudi Arabia	DB, RD, CT	T2DM, 30–60 y 30/31	sachet	*B. bifidum* W23, *B. lactis* W52, *L. acidophilus* W37, *L. brevis* W63, *L. casei* W56, *L. salivarius* W24, *Lactococcus lactis* W19, *Lactococcus lactis* W58 2.5 × 10^9^ CFU/g	24 wk	S↓ FG,	S↓ I, HOMA-IR,	S↓ TC, TG, total/HDL-cholesterol ratio	-
[173]	2019	Iran	parallel-group, RD, CT	T2DM, 20/20 30–50 y	capsule	*L. casei,* 10^8^ CFU/d	8 wk	S↓ FG =HbA1c	S↓ I, HOMA-IR	-	-
[174]	2019	Iran	RD, DB, CT	T2DM 30–75 y 34/34	capsule	*L. acidophilus,* 2 × 10^9^ CFU *L. casei,* 7 × 10^9^ CFU *L. rhamnosus,* 1.5 × 10^9^ CFU *L. bulgaricus,* 2 × 10^8^ CFU *B. breve,* 3 × 10^10^ CFU *B. longum,* 7 × 10^9^ CFU *S. thermophilus,* 1.5 × 10^9^ CFU 100 mg FOS	6 wk	S↓ FG	=I, HOMA-IR	S↑ HDL-C =TG, TC	-
[175]	2019	India	RD, DB, CT	T2DM, Ob 18–65 y 39/40	capsule	*L. salivarius*, *L. casei*, *L. plantarum*, *L. acidophilus*, *B. breve*, *B. coagulans*, 30 billion CFU 100 mg FOS	12 wk	S↓ HbA1c =FG	=I, HOMA-IR	= TC, TG, HDL-C, LDL-C	-
[133]	2019	Belgium	RD, DB, PC, pilot study	Ob/ov insulin-resistant volunteers 18–70 y 14/13/13	sachet	Live/pasteurized *Akkermansia municiphila* 10^10^ bacteria/day	12 wk	=FG, HbA1c	S↑ insulin sensitivity S↓ I	S↓ TC =LDL-C, TG	=GLP-1
[176]	2020	Australia	RD, DB, CT	T2DM BMI ≥ 25 kg/m^2^ ≥ 18 y 30/30	capsule	*L. plantarum,* 6 × 10^9^ CFU, L. *bulgaricus,* 3 × 10^9^ CFU L. *gasseri,* 18 × 10^9^ CFU B. breve, 7.5 × 10^9^ CFU B. animalis sbsp. lactis, 8 × 10^9^ CFU B. *bifidum,* 7 × 10^9^ CFU *S. thermophiles,* 450 × 10^6^ CFU *Saccharomyces boulardii,* 45 × 10^6^ CFU	12 wk	S↓ FG, HbA1c (in patients taking probiotics and metformin)	S↓ HOMA-IR (in patients taking probiotics and metformin)	-	-

RD, randomized; DB, double-blind; PC, placebo-controlled; SB, single-blind; CT, clinical trial; T2DM, type 2 diabetes mellitus; Ob, obese; NASH, Nonalcoholic steatohepatitis; Ov, Overweight; NAFLD, Non-alcoholic fatty liver disease; MS, Metabolic syndrome; GDM, gestational diabetes mellitus; CHD, coronary heart disease; L., Lactobacillus; B., Bifidobacterium; S., Streptococcus; CFU, colony-forming units; FOS, fructooligosaccharides; wk, weeks; d, days; FG, fasting glucose fasting blood glucose, fasting plasma glucose, glycemia, fasting blood sugar; HbA1c, Hemoglobin glycated; OGTT, glucose tolerance test; I, serum insulin concentration, insulin concentration, serum insulin level, insulin; HOMA, homeostatic model assessment; HOMA-IR, homeostasis model of assessment-estimated insulin resistance; QUICKI, quantitative insulin sensitivity check index; HOMA-B, homeostasis model assssessment of β-cell dysfunction; TG, triglycerides; VLDL, very-low-density lipoprotein; TC, total cholesterol; HDL-C, high-density lipoprotein; LDL-C, low-density lipoprotein; NEFA, non-esterified fatty acids; Apo B-48, apolipoprotein B-48; oxLDL, oxidatively modified low density lipoprotein; GLP-1, Glucagon-like peptide-1; GLP-2, Glucagon-like peptide-2; =, non significan; S, significant; ↑, increase; ↓, decrease.

## Data Availability

Not applicable.

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
