# Peer review of "Gut Microbiota and Complications of Type-2 Diabetes"

_nutrients, 2021, doi:10.3390/nu14010166_

Round 1

Reviewer 1 Report

The manuscript needs moderate english resvisions. The other materials looks good. 

Author Response

We appreciate the positive review. We have revised the paper with an eye toward correcting spelling errors and other syntax construct.

Reviewer 2 Report

  Share

 Dear Sirs, this is a very interesting manuscript dealing witha very uptodate issue. It is very well-written and justified. However, in my opinion, the following reference should be added, as it is the only study with the addition of Akkermanisa muciniphila in humans until now.

Supplementation with Akkermansia muciniphila in overweight and obese human volunteers: a proof-of-concept exploratory study. Depommier C, Everard A, Druart C, Plovier H, Van Hul M, Vieira-Silva S, Falony G, Raes J, Maiter D, Delzenne NM, de Barsy M, Loumaye A, Hermans MP, Thissen JP, de Vos WM, Cani PD.Nat Med. 2019 Jul;25(7):1096-1103. doi: 10.1038/s41591-019-0495-2. Epub 2019 Jul 1. In addition, the problems with Akkermanisa muciniphila and Faecalibacterium prusnitzii as next generation probiotics should be added in a section, too, from my point of view.

Author Response

Thank you for your constructive evaluation, which helped improved our paper. As suggested, we have added the main findings from Depommier C. et al. paper and briefly addressed the emergence and importance of next generation probiotics, with specific emphasis on the effects of Akkermanisa muciniphila and Faecalibacterium prusnitzii (Line 408-425).

The added text reads as follows:

For example, in a proof-of-concept, randomized double-blind controlled clinical trial study, Depommier et al. showed that supplementation for 3 months with A. muciniphila significantly improved insulin sensitivity, reduced insulinemia, plasma total cholesterol and inflammation [133]. These results show that intervention with specific bacteria strains may prove a useful strategy in improving metabolic parameters associated with diabetes and its complications. Indeed, several bacteria with enhanced functional characteristics in treating specific host diseases have been defined as next generation probiotics (NGP). Among them, Akkermansia muciniphila, Ruminococcus bromii, Faecalibacterium prausnitzii, Anaerobutyricum hallii and Roseburia intestinalis have gained considerable interest and have been the primary candidates. In particular, A. muciniphila have been associated with improved metabolic endotoxemia, amelioration of metabolic syndrome phenotype, improved lipid and glucose metabolism and may serve as diagnostic tool for dietary interventions. Likewise, Faecalibacterium prausnitzii has been shown to exert anti-inflammatory action and has been proposed as a biomarker for the development of gut diseases and for assessing dietary interventions in intestinal inflammatory conditions [134] (Table 2). Based on these findings, several novel food and pharma supplements have been developed with profound beneficial effects in protecting from specific metabolic disorders and other metabolic risks.